## Article

# Philosophy of Religion in a Fragmented Age: Practice and Participatory Realism

**Jacob Holsinger Sherman**

Philosophy and Religion Department, California Institute of Integral Studies, San Francisco, CA 94103, USA; jsherman@ciis.edu

**Abstract:** What should the philosophical study of religion look like in an epoch of increasing political polarization, cultural ferment, and religious fragmentation? Drawing on the work of Amy Hollywood and others, I argue that philosophers seeking to understand what seem to be incommensurable moral and religious communities ought to attend more fully to the role of spiritual practice and moral formation as irreducible components of certain beliefs and ethical intuitions. However, while such an account might invite a reductive reading in which the object of religious belief is taken to be simply the practice, ritual, etc., I engage the thought of Michael Polanyi to argue that such irreducibly participatory truth claims can be understood to aim at a reality that exceeds the structures of formation and ways of life to which they are indexed.

**Keywords:** ritual; participation; spiritual practice; participatory realism; Polanyi; participatory epistemology

What should the philosophical study of religion look like in an epoch of increasing political polarization, cultural ferment, and religious fragmentation? One approach for philosophers of religion thinking about matters of justice, morality, and ethics is to attend to the religious beliefs that support and mobilize the ethical claims of various groups. How, for example, does *pratityasamutpada* motivate the extension of ethical circles to nonhuman kin in Plum Village? How do Christological claims motivate the vision of the dignity of the person in Catholic social teaching? There is a tremendous amount to commend in such approaches insofar as what we explicitly believe to be the case—including what we believe to be religiously the case—plainly shapes what we consider it right to do; therefore, bringing these beliefs to reflective articulation plainly aids our efforts at ethical deliberation. However, in an age marked by polarization, such an approach also risks exasperating the sense that we live amidst incommensurable moral communities rooted in fundamentally different first principles. As Charles Taylor has argued, in the face of such pluralism, our deeper moral reflection needs to go beyond the question of 'what it is right to do' in order to ask also 'what is it good to be?' ([Taylor 1989](#), p. 3). For Taylor, answering that question means, among other things, uncovering the implicit moral ontologies integral to one's deepest moral responses and intuitions, on the one hand, and providing a rich account of their historical emergence, on the other. Beyond this genealogical aspect, though, we might also ask how one comes to live in such moral and spiritual worlds at all. This, in turn, might help us move beyond the problem of incommensurability by facilitating conversations that need not stop at rival moral intuitions and competing religious claims but could possibly carry on into convivial accounts of how such competing claims could be seen and felt to be true.

In order to do this, the philosophical study of religion would have to attend at least as much to practice as to doctrine, to processes of formation as much as propositional claims. Here, the philosophy of religion, with its focus on the coherency and adequacy of religious beliefs and frameworks, on the one hand, might have much to learn from religious studies with its focus on thick, regional descriptions of religious communities, on the other ([Lewis 2015](#); [Schilbrack 2014](#); [Trakakis 2008](#); [Wainwright 1996](#)). The estrangement

of these two guilds from one another has been unfortunate for both disciplines: a purely historical and social scientific turn in religious studies leaves critical questions about the ontology of religious experience and belief unaddressed and effectively cedes the ability to mount normative religious claims, while philosophers have too often proceeded as if the properly epistemological and metaphysical dimensions of our inquiries could be separated from accounts of their embeddedness in religious ethics, practice, ritual, community, and formation. Arguably, if we are to mobilize our profession to better address the tears in our social fabric, we will need a more integral philosophical study of religion, one capable of better attending to the entwining of lived practice with metaphysical and ethical claims.

An example might help to make this more concrete. Consider the following claim drawn from Bonaventure's *Collations on the Hexaemeron* 12.14:

> . . . the whole world is a shadow, a way, a vestige, and it is a book written on the outside. For in whatever creature, there is a reflection of the Divine Exemplar (*refulgentia divini exemplaris*), although mixed with darkness; hence it is like a certain opacity combined with light (*lumini*). Likewise [the sensible creature] is a way leading to the Exemplar. Just as you see that a ray of light (*radius*) entering through a window is variously colored according to the different colors of the diverse parts; so the divine ray of light (*radius divinus*) is diverse in individual creatures and it reflects in diverse properties; in Wisdom: [Wisdom] shows herself to them . . . in the ways. Likewise [the sensible creature] is a vestige of the wisdom of God. Hence the creature is nothing other than a certain representation (*simulacrum*) of the wisdom of God, like a type of figure (*simulacrum*). And for all of these reasons, it is a kind of book . . . (Bonaventure 2018, pp. 281–82)

Faced with a passage such as this, scholars of religion, aware that religious practices and the content of religious confessions may vary from society to society and person to person, might be led to ask about the relation between medieval urbanization and the Franciscan focus upon incarnation, Bonaventure's fusion of the nature mysticism of Francis with the soul mysticism of Augustine and Bernard of Clairvaux, or about the appearance of *Sapientia* as an instance of that 'medieval goddess piety' that runs from Boethius through Alan of Lille and Hildegard of Bingen, to Geoffery Chaucer and Spenser's 'Mutabilitie Cantos'. The philosopher, by contrast, might perceive a set of more putatively timeless questions about the possibility of natural theology, the metaphysics of divine ideas, divine wisdom as a regulative ideal, the epistemological aspects of Augustinian illumination, or the ontology of created beings. The questions, however, that would fall in between these disciplinary divides may be the most interesting and morally consequential having to do with whether and how the world can be read in this manner and what sorts of responses are appropriate within such a world. This is where questions of moral and spiritual formation become salient. Is the real world itself susceptible to being encountered as a book, such that finite creatures are perceived as symbols diaphanous to the Logos? If so, how is such an encounter possible? How do we judge the reality of such putative disclosures? And how does this change the texture of our life together with human and non-human creatures, including that normative form of togetherness we call justice? To address questions of this sort, we need something more than either the social scientific and historical study of religion, on the one hand, or a purely cognitive approach to the philosophical study of religion, on the other.

Earlier in the *Collations* (1.17), Bonaventure writes, "this is the whole of our metaphysics: emanation, exemplarity, and consummation, that is, to be illumined (*illuminari*) by spiritual rays of light (*radius spirituales*) and to be led back (*reduci*) to the Most High (*summum*). And in this way, you will be a true metaphysician." (Bonaventure 2018, p. 106) This last element—this illumination and consummation leading back to the absolute—points away from a vision of metaphysics as a purely abstract, cognitive exercise and to a kind of metaphysics suffused with soteriology and spiritual formation, a vision that will be understood only if we take into account both Bonaventure's philosophical claims and the practices and realizations to which those claims are ineluctably bound.

What might it look like for philosophers of religion to take practice seriously? In 'Practice, Belief, and Feminist Philosophy of Religion,' Amy Hollywood, herself a philosopher and scholar of medieval mysticism, argues that moving beyond what she calls the cognitivist bias of contemporary philosophy of religion requires that philosophers turn their attention to the central place of ritual, formation, and practice within religious life and confession (Hollywood 2016, pp. 233–50). To be sure, throughout the humanities, the concept of practice has become quite popular, but where metaphysics and religion are concerned, this new interest in practice is often interpreted in a decisively post-Kantian manner. According to this Kantian view, the true objects of religious language are neither gods and goddesses nor sephirotic emanations or the wisdom hidden in the heart of all things, but certain regulative ideals towards which human cognition and volition aim but from which no constitutive knowledge of the super-sensible can be drawn. Where Kant saw such ideals as the consequence of the transcendental shape of pure and practical reason, more recent works tend to read such regulative ideals in a kind of Durkheimian manner, as both an expression and determinant of one's culture, class, society, history, language, and so forth. Here, Hollywood cites the work of the feminist philosophers of religion, Pamela Sue Anderson and Grace Jantzen, as examples of this post-Kantian tendency. Although, formally, Anderson writes more from the analytic tradition and Jantzen from the Continental, both projects concern themselves materially with a constructive philosophical account of the feminine divine. Anderson aims to defend a kind of realism in feminist speech about God by appealing to the broadly Kantian sense that the objects of theological language are certain regulative ideals towards which human cognition and volition aim but from which no constitutive knowledge of the super-sensible can be drawn. Beyond Kant, Anderson argues for something akin to a regulative ideal of desire directed towards a vision of moral and political flourishing. As Anderson writes, "For feminist philosophy, authentically conceived and strongly objective theistic beliefs of women would not come from psychological need alone, nor from epistemological ignorance but, significantly, from a rational passion for justice" (Anderson 1998, p. 213).

For her part, Jantzen also follows the route of a modified Kantianism, albeit filtered through a robust engagement with the thought of Ludwig Feuerbach. For Jantzen, theological language is the language "of ideals, indeed regulatory ideals in a Kantian sense. Projections need to be those which embody our best and deepest aspirations, so that we are drawn forward to realize them" (Jantzen 1999, p. 92). Even on these Feuerbachian terms, there are problems with such a project. Most notably, the self-consciousness of it all renders it deeply suspect, for Feuerbach held that any rationally generated projection deliberately deployed as a projection could only function as an abstraction—an allegory at best—and would exert perforce little persuasive power. What Ernst Cassirer wrote about Schelling's philosophy of mythology applies equally to Feuerbach's philosophy of religion:

> the phenomenon which is here to be considered is not the mythical [or theological] content as such but the significance it possesses for human consciousness and the power it exerts on consciousness. The problem is not the material content of mythology, but the intensity with which it is experienced, with which it is believed—*as only something endowed with objective reality can be believed*. (Cassirer 1953, p. 5, emphasis mine)

In this manner, the potency of a religious projection issues directly from its having been engendered by imagination and desire apart from rational control. Naturally, the Romantic Platonists among us will insist that, at their highest, rationality, imagination, and desire cannot be separated. However, Feuerbach's argument is that, pragmatically, it is only as separated from reason that desire and imagination can generate the powerful projections of religious faith, for only then is the object of belief (mis)understood to be ontologically independent of the believer.

The modern temptation is to treat the turn to practice as just another chapter in this Kantian story, following the early-modern turn to the subject and the late-modern linguistic turn (Sherman and Ferrer 2008). Hollywood's challenge, however, is more sophisticated:

it is not that one ought to turn to practice in order to dispense with questions about the ontology of the abundant or hyperbolic real and of those experiences deemed religious, but rather that one can only ask the philosophically interesting questions about the normative and ontological status of religious claims after one has sufficiently interrogated the place of practice, ritual, and moral formation in the constitution of the discursive, symbolic, and mythic productions of religious life.

As far as the philosophical study of religion is concerned, the problem with these post-Kantian approaches is that they abandon too quickly and too confidently the religious, ethical, and metaphysical realism of ordinary believers and so they elide "what people most often mean when they say that they believe in God." (Hollywood 2016, p. 237) Before we offer functionalist reinterpretations of religious practices and claims or apologetic justifications of the same, we need to at least understand what they really mean to those making them. "From this perspective," Hollywood argues:

> the question of the ontological status of the objects of religious belief cannot be resolved by redefining the nature of divine existence in terms of regulative ideals (be they epistemological or moral). Nor can religion be equated solely with belief. For many religious people, practice takes precedence over belief; a philosophy of religion that does not account for the function and meaning of practice will never be adequate to its object. (Hollywood 2016, p. 238)

Her insinuation here is overstated: philosophers have long known that practice, and the moral and spiritual formation to which it is indexed, is central to religious forms of life, but the challenge has always been how to make philosophical sense of this centrality without indulging in some form of contrastive nonrealism. Important studies by Peter Ochs, Catherine Pickstock, and more recently Nicholas Wolterstorff have sought to address this challenge, as has much of Hollywood's own writing (Ochs 2006; Pickstock 1998; Wolterstorff 2018). Hollywood's most important claim is that taking practice seriously leads to the discovery that practical reason is not a given that can accordingly be universalized in a Kantian manner; rather, practical reason is constructed through "learned modes of being in the body and the world." (Hollywood 2016, p. 241) In this regard, she appeals to Talal Asad's account of ritual practices as acts of formation. Asad argues against those who treat ritual as secondary, a repetitive and coded behavior whose real meaning lies in the social institutions to which it points (Asad 1993, pp. 48–50). On such an account, it falls more to the anthropologist, philosopher, or sociologist than to the practitioner to uncover a ritual's true import; the mysteries of the academy give the scholar a kind of esoteric insight into the ritual's true nature, an insight prohibited to those who merely participate in the rites themselves. Against this modern understanding of ritual, Asad points to medieval monastic accounts of ritual as disciplinary practice. As Asad notes, this monastic understanding of practice can be summed up in Hugh of Saint Victor's conception of "ritual gesture and speech as the discipline of the body that is aimed at the proper ordering of the soul" (Asad 1993, p. 139). In this context, ritual no longer appears as an activity whose content needs to be indexed to some additional, more explanatorily basic category, but is understood as what it seems to be: a praxis that shapes one's moral life, aids in the acquisition of the Christian virtues, and furthers the flourishing of the community in communion with God and all creation.

> The formation/transformation of moral dispositions . . . required a particular program of disciplinary practices. The rites that were prescribed by that program did not simply evoke or release universal emotions, they aimed to construct and reorganize distinctive emotions—desire (*cupiditas/caritas*), humility (*humilitas*), remorse (*contritio*)—on which the central Christian virtue of obedience to God depended. This point must be stressed, because the emotions mentioned here are not universal human feelings, not 'powerful drives and emotions associated with human physiology'... They are historically specific emotions that are structured internally and related to each other in historically determined ways. And they

are the product not of mere readings of symbols but of processes of power. (Asad 1993, p. 134)

Ritual practices here appear as acts of formation, practices through which subjectivity, affect, virtue, and belief are cultivated and habituated—leading to what Marcel Mauss, following Aristotle, called *habitus* (Asad 1993; Mauss 1979).

Pushing now beyond Hollywood and Asad—though not, perhaps, beyond Hugh of Saint Victor—we might wonder whether it is not only practical reason but even theoretical reason that could be so transformed. If ritual is a practice of formation through which one acquires new capacities of body, soul, and spirit, could it be that certain beliefs, moral intuitions and spiritual realizations are not merely speculative inferences from some putatively neutral and publicly observable world but are the skillful achievements of disciplined bodies and souls? As Hollywood herself provocatively suggests: "If mystical states are understood as God's existence made bodily and spiritually inescapable, then this leads to the possibility that skepticism about or disbelief in God's existence is itself, as Talal Asad puts it, 'a function of untaught bodies'" (Hollywood 2016, p. 239).

There is a self-implicating quality to these arguments that runs against the epistemological and political presuppositions of philosophical liberalism, suppositions that depend on a strict bifurcation of nature and culture and thus upon a constitutive forgetting of the acquired nature of one's own habitus. Hollywood recognizes the danger in all of this but thinks it worth courting insofar as it permits us to move forward in our understanding of others, and especially in our understanding of what seem to be intractable debates otherwise that conclude to unmediable competing basic intuitions. Picking up where Hollwood leaves off, we might argue along similar lines for the recognition of doxastic conditions that include spiritual practice and moral formation as irreducible components of certain beliefs and ethical intuitions. Written in declarative sentences, such beliefs might look akin to ordinary propositions, but the key point is that some of these propositions can only be countenanced within a wider semantic field composed of correlative habits. Let us call these habitus-indexed propositions *participatory truth claims*. A participatory truth claim is a claim that cannot be properly entertained by untaught bodies and minds, which is to say outside a certain *paideia*.

Now, having arrived at a thick description of religious practices and their role not only in religious belief but also in the metaphysical and moral claims implicit in religious belief, it might be tempting to say that the object of religious belief simply *is* the practice, ritual, or habitus itself. This post-metaphysical option, however, would merely reinscribe at a deeper, more hidden level precisely the bracketing of ontology that was already found to be problematic. The question for philosophers of religion is whether such irreducibly participatory truth claims can be understood to aim at a reality that exceeds the structures of formation and ways of life to which they are indexed.

For the philosophical study of religion, this question ought to be of central concern, for religions at their most basic have always sought to put their adherents into contact—not only with a trope or story or regulative ideal—but also with a reality so real that it shapes, qualifies, orders and contextualizes the more quotidian realities within which one otherwise ordinarily lives. Premodern traditions, such as those Bonaventure and Hugh of Saint Victor inherited, take it for granted that one's capacity for intellectual or contemplative insight grows as a result of an integral education through which one learns to be in right relation to the sensory, moral, intellectual, and spiritual worlds to which one belongs, as well as those transcendent realities that perhaps give all such worlds to be. However, more contemporary iterations of this sort of participatory realism can also be defended.

The tradition of post-Heideggerian hermeneutics, for instance, from Hans Georg Gadamer to Paul Ricoeur to Charles Taylor, has insisted on the importance of attending to the way that the human mode of existence is constituted by meanings that are themselves shaped by our ongoing efforts at self-interpretation. At the outset of *Truth and Method*, for example, Gadamer argues that the humanities properly aim not at an *objective knowing freed from subjective points of view,* but rather at the shaping of human beings capable of coming to

ever greater understandings of one another, of our texts and artifacts, and of ourselves. As Gadamer writes, "What makes the human sciences into sciences can be understood more easily from the tradition of the concept of *Bildung* than from the modern idea of scientific method" (Gadamer 2004, p. 16). Initially associated with culture—and the acquisition of culture—*Bildung* came to refer to something more akin to a process of spiritual formation, what I called above a *padeia*. "The rise of the word Bildung," writes Gadamer, "evokes the ancient mystical tradition according to which man carries in his soul the image of God, after whom he is fashioned, and which man must cultivate in himself. The Latin equivalent for *Bildung* is *formatio*, with related words in other languages—e.g., in English (in Shaftesbury), "form" and 'formation'" (Gadamer 2004, p. 10).

For Gadamer, the humanities are constituted by the aim of being initiated into traditions that allow us to read and understand the art and literature that preceded us. Indeed, we have no other way to understand history itself for history is governed not by exceptionless laws of nature that we might discover through the right use of a method, but by the moral realm of freedom and the mystery of the person. "For this reason," Gadamer writes, "historical research does not seek knowledge of laws and cannot appeal to the decisiveness of experiment. For the historian is separated from his object by the infinite mediation of tradition" (Gadamer 2004, p. 213). However, against post-Enlightenment readings of tradition and prejudice as stumbling blocks on the way to truth, Gadamer sees tradition as truth-conducive, at least within the realm of the humanities. As Gadamer puts it, "this distance is also proximity". Why? What is it that allows tradition to bring us into deeper forms of experience and understanding? Ongoing initiation into a tradition brings one into contact with the meaningfulness of the past through the medium of our own transformed subjectivity. "The historian does not investigate his 'object' by establishing it unequivocally in an experiment," writes Gadamer. "Rather, through the intelligibility and familiarity of the moral world, he is integrated with his object in a way completely different from the way a natural scientist is bound to his" (Gadamer 2004, p. 213).

In the hermeneutic tradition, then, we find an account of philosophy and of knowledge, in general, that takes formation seriously as constitutive of our capacity to know truth; however, if participatory truth claims are limited to the human sciences in the manner described by Gadamer, then philosophers of religion may find themselves disappointed once again. The post-Heideggerian hermeneutical approach—so powerfully described by Gadamer and developed in other ways by philosophers such as Merleau-Ponty and Charles Taylor—rightly recognizes that knowledge about the moral realm, about history, and about persons requires the formation of bodies and minds in certain tradition-inflected ways. However, if this approach applies only to the realm of culture and not also, in some ways, to nature, then we seem to be caught in another iteration of the aporetic ontological bracketing described above.

For this reason, I want to conclude by considering the even more radical approach of the Hungarian chemist-turned-philosopher Michael Polanyi, who argues powerfully for *the epistemic centrality of personal participation* as integral to *all* knowledge of the real. Polanyi spent the first half of his career as a world-class scientist himself and knew that the processes of discovery involve a fundamental act of imagination through which the scientist deploys his or her own subjectivity "as the principal link" with the world's intelligibility or "subjectivity." (Polanyi and Prosch 1975, p. 57) As Polanyi writes, "We now see that not only do the scientific and the humanistic both involve personal participation; we see that both also involve an active use of the imagination" (Polanyi and Prosch 1975, p. 64). Nevertheless, the sources of imagination and inspiration are ineluctably personal and thus resist any final formalization. Therefore, Polanyi argues, if we are to understand how discovery takes place—in other words, if we are to understand how we come to know those real but hitherto unapprehended dimensions of the world at the heart of science, ethics, aesthetics, and faith—we need to broaden our inquiry to include all those diverse processes of formation, training, and practice through which the person, the primary locus of discovery in all dimensions, comes to be (Polanyi 1958, 1966). Although the personal

coefficient of knowledge cannot be formulated, Polanyi provides a sophisticated account of the way in which the person herself is formed and transformed through education, apprenticeship, and the acquisition of new habits, skills, and tools.

The transformation of the person opens new horizons of knowledge, for we know the world only through integral, imaginal acts of personal participation or indwelling. One of Polanyi's ways of addressing this participatory aspect is to speak of the 'tacit dimension' in both scientific and everyday knowing. Tacit knowledge begins from the fundamental insight that "we know more than we can tell" (Polanyi 1966, p. 4). As with Bergson, Heidegger, or Whitehead before him, Polanyi rightly takes tacit knowledge to be epistemically primary, and considers our more explicit, articulate acts of knowing to be secondary, higher-order affairs. It is not only the case that we know more than we can tell but also that "we can tell nothing without relying on our awareness of things we may not be able to tell" (Polanyi 1958, p. x).

Tacit knowing is itself composed of two levels of awareness: a focal awareness of the whole (the *molar* level) and a subsidiary awareness of parts (the *molecular* level). When we attend directly to anything—a cello suite, a vegetable garden, a spike protein, the presence of Christ in a stranger, the suffering of sentient beings in Tonglen meditation—we become aware of it focally. However, this focal awareness is only made possible by the subsidiary awareness we have of its parts. The parts that we perceive subsidiarily act as clues leading us towards the comprehensive entity that we know focally (Polanyi 1969). That said, how do we move from the subsidiarity of the parts to a vision of the whole? This process is only accomplished through the personal or tacit integration of the parts. This personal integration, which Polanyi regularly describes in the language of Gestalt theory, is not a *merely subjective act* for it follows the clues given to it by the molecular reality in our subsidiary awareness. In other words, the tacit integration that alone reveals a meaningful world is accomplished only insofar as the person feels his or her way into something akin to the grain of the universe. By indwelling the parts and, as it were, reading their own aspirations, the person becomes the site for the emergence of meaningful wholes.

> The knowledge of a problem is, therefore, like the knowing of unspecifiables, a knowing more than you can tell. But our awareness of unspecifiable things, whether of particulars or of the coherence of particulars, is intensified here to an exciting intimation of their hidden presence. It is an engrossing possession of incipient knowledge which passionately strives to validate itself. (Polanyi and Grene 1969, pp. 131–32)

Knowledge, on this account, is less a matter of representation than a relational achievement. He even goes so far as to say that scientific discovery may be better understood as a process "guided . . . by an aspect of nature seeking realization in our minds"(Polanyi 1946, p. 35).

Mediation, rather than representation, is primary in the act of knowing. To know anything, as finite beings, we rely on a series of mediations; crucially, however, these mediations are not separations, for we are capable of *indwelling* these intermediaries and therefore able to come through them to real communion with that which is known. The paradigm for such indwelling is our bodies; this is what makes our bodies cosmically unique. We attend *from* our bodies *to* the meaningful world in which our bodies themselves participate. There is a continuum or polarity that holds between the somatic and the conceptual, one that can be mapped to the continuum of the subsidiary and the focal. Polanyi speaks of the 'exceptional position' of the body in the universe:

> [T]he way the body participates in the act of perception can be generalized further to include the bodily roots of all knowledge and thought. Our body is the only assembly of things known almost exclusively by relying on our awareness of them for attending to something else. Parts of our body serve as tools for observing objects outside and for manipulating them. Every time we make sense of the world, we rely on our tacit knowledge of impacts made by the world on our body and the complex responses of our body to these impacts. (Polanyi and Grene 1969, p. 147; Polanyi and Prosch 1975, p. 36)

The body is thus the first and most intimate instance of the "from–to" structure of knowing that is central to Polanyi's account of personal knowledge. In a manner similar to that of Varela, Thompson and Rosch (Varela et al. 1991), Polanyi insists that what is most our own in knowledge is our bodily awareness, while the intellective pole of our knowledge is fundamentally the knowledge of the intelligible world in which our bodies are immersed.

Our bodies are only defined as ours by the fact that we do indeed indwell them, but we prove capable of indwelling much more besides. When we use a probe, for example, it initially strikes us as foreign and other, but as we integrate the probe into our subsidiary awareness, our focal attention moves from the probe itself to the impact that the end of the probe makes on the area it explores. In this way, through practice, the probe becomes an extension of our body and begins to disclose the intelligibility of the world to us, which is also to say, it allows the intelligibility of the world to begin to answer us, even to shape us. We might even speak of a kind of ecstatic transcorpeality at work here—without ever taking leave of the somatic, Polanyi's theory demonstrates the ecstatic capacity of our bodies through training to exceed their skin-encapsulated bounds; we discover what is other than us and discover ourselves changed, expanded, and made-other in the process.

In his remarkable discussion of intellectual passions, Polanyi extends this concept of indwelling to our articulate frameworks themselves. These frameworks thus become less like conceptual schemes that prevent us either from understanding one another or from touching the world in its own reality, and more like tools or probes, parts of the world themselves that we indwell in order to make contact with a world that grows ever larger. As Polayni writes, "A valid articulate framework may be a theory, or a mathematical discovery, or a symphony. Whichever it is, it will be used by dwelling in it, and this indwelling can be consciously experienced" (Polanyi 1958, p. 208). Religions, too, as comprehensive and personally transformative ways of life, can be similarly indwelt. At its limit case, Polanyi notes, this personal indwelling involves something akin to what the 'religious mystic' experiences or enacts in contemplation.

> By concentrating one's focal awareness on the presence of God, who is beyond all physical appearances, the mystic seeks to relax the intellectual control which his powers of perception instinctively exercise over the scene confronting them. His fixed gaze no longer scans each object in its turn and his mind ceases to identify their particulars. The whole framework of intelligent understanding, by which he normally appraises his impressions, sinks into abeyance and uncovers a world experienced uncomprehendingly as a divine miracle. (Polanyi 1958, p. 197)

Although such contemplation is often spoken of as the achievement of detachment or *apatheia*, there is nothing here to correspond to the scientific ideal of impersonal detachment and putative objectivity. Rather, "the impersonality of intense contemplation consists in a complete participation of the person in that which he contemplates and not in his complete detachment from it, as would be the case in an ideally objective observation . . . ." (Polanyi 1958, p. 197). Accordingly, going a bit beyond Polanyi, we could speak of this contemplative indwelling as the formation or evocation of the *contemplative body*. The successive detachments of the contemplative from objects that are usually perceived focally are not in themselves final, as if pure asceticism were the goal. In one sense, the contemplative never really detaches at all; instead, he or she integrates all these elements into subsidiary awareness so that the he or she can attend to the hyperbolically excessive horizon of the world, even to God who meets the contemplative in-and-beyond the world. Akin to the example from Bonaventure at the beginning of this essay, Polanyi explains that the contemplative endeavors "through a succession of 'detachments', to seek in absolute ignorance union with Him who is beyond all being and all knowledge. We see things then not focally, but as part of a cosmos, as features of God" (Polanyi 1958, p. 208).

Whether in philosophy, science, or religious life, we know the natural and the more-than-natural worlds within which we live only to the extent the we attend from our subjectivity to these worlds through a continuum of media that become ours through practice and familiarization: from the bodies that each of us as infants must learn to

negotiate to the cascade of material objects, languages, idioms, cultures, rituals, concepts, imaginative worlds, intellectual passions, aesthetic ideals, meditative disciplines, and so forth. By attending to these diverse pathways of formation, the philosopher of religion is able to show how we come to dwell in communities whose apparent incommensurability need no longer stop conversation, but can instead become invitations to better understand how diverse participatory truth claims are made intelligible through the acquisition of particular habits and forms of life. Crucially, however, there is nothing here that should encourage critical skepticism about the possibility that these participatory claims may involve real metaphysical or religious knowledge. Where critical philosophy seeks both to distinguish and to distance the knower from the known, Polanyi's project is better characterized as participatory and post-critical. For the participatory philosopher, the personal and subjective coefficient of all knowing does not erect a barrier to knowledge of the world; rather, realism is possible because the world itself is seen as implicitly personal. As Stephen R L Clark writes, "Scientists, like more traditional believers, reckon theses 'true' if they agree, not simply with whatever we already believe, but with the world . . . . If our words and the world agree there must be a sense in which the world actually says something and is therefore 'personal'!" (Clark 1991, p. 4).

If this is the case, then we have even more reason to hope that our polarization and fragmentation need not be the last word. Not only can we come to understand how diverging *habitus*-indexed claims can be sincerely held by varying communities ordered to what they experience as the lure of the good, but we can also hope that our differences may ultimately be overcome insofar as the participatory knowledge of the world is still knowledge, and aims therefore at that which is public, even at the universal. The philosopher of religion can affirm this because the philosophical study of religion need not choose between a kind of abstract cognitivism, on the one hand, and a post-metaphysical historicism on the other, for there is no transcendental gulf separating nature and culture, the rational and the real. Rather, the frameworks that we acquire through apprenticeship, moral formation, religious practice, ritual participation, and all the various adventures of life are less akin to conceptual schemes that prevent us from touching the world in its own reality, and more similar to tools or probes, parts of the world themselves that we indwell in order to make contact with what Polanyi calls "the subjectivity (the hidden lawfulness) of the world" (Polanyi and Prosch 1975, p. 57).

**Funding:** This research received no external funding.

**Institutional Review Board Statement:** Not applicable.

**Informed Consent Statement:** Not applicable.

**Data Availability Statement:** No new data were created or analyzed in this study. Data sharing is not applicable to this article.

**Conflicts of Interest:** The author declares no conflict of interest.

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
