# Peer review of "Philosophy of Religion in a Fragmented Age: Practice and Participatory Realism"

_religions, doi:10.3390/rel14030424_

Round 1

Reviewer 1 Report

The idea behind the project is interesting. The problem is that the article takes too many basic principles for granted. Furthermore, although the authors cited are relevant, a more complete discussion of the topics mentioned is required. A different structure of the text, divided into paragraphs, could support understanding of the contents.

Author Response

Please check the revised version.

Reviewer 2 Report

Line 100: The leap of thought from Bonaventure to Amy Hollywood (2016) would require a clearer argumentative bridging.

Line 144: The way of putting arguments by authors from distant time periods is a weakness of the theoretical foundations of the research (e.g. Cassier, 1953).

Line 167: The argument should be a suitable starting point for profiling the research purpose in the empirical part.

Lines 230-237: He follows up on Amy Hollywood (2016) by emphasizing the impact of teaching on religious belief, which gets to the heart of his argument.

Line 256: The reference to Polanyi (1966) includes an important argument for the current understanding of cognition.

Lines 343-376: By analyzing the text from Polanyi (1958), author returns to the topic of religious experience.

Author Response

Please check the revised version.

Reviewer 3 Report

This is a clear and persuasive article that I enjoyed reading very much. I also benefitted from the unique constellation of literature (e.g. Hollywood, Polanyi), with which the author engaged. The structure and argumentation are solid. 

My recommendation is to accept this article in its present form. 

The one suggestion I would make--more for further consideration rather than any necessary revision of this article--is to explore those other thinkers who have been engaged, similarly, in bridging the gap between the philosophy of religion and religious studies. I'm thinking especially of the hermeneutic tradition that runs from Gadamer to James K. A. Smith (whom I mention only because Wolterstorff is cited here also). I would welcome reading more about how Polanyi, as well as Hollywood and others, might fit into this larger tradition of mediating philosophy and history, idealism and cultural studies, beliefs and practice.

Good work has a way of producing more good work to be done. So, thank you for your good work on this article. 

See attached document for more detailed feedback. 

Author Response

Please check the revised version.